

# Optimizing output operations in high-resolution climate models through dynamic scheduling

Dong Wang[1], Xiaomeng Huang[1]

[1] Department of Earth System Science, Ministry of Education Key Laboratory for Earth System Modelling, Institute for Global Change Studies, Tsinghua University, Beijing 100084, China

*Correspondence to*: Xiaomeng Huang (hxm@tsinghua.edu.cn)

**Abstract.** This study presents a new approach to improve the efficiency of data output in high-resolution climate models. The method begins by forwarding data to processes with lighter workloads or finishing their tasks earlier, allowing these units to serve as temporary storage. Following this, the processes create multiple smaller communication groups to reorganize the data and then use an I/O aggregation approach to enable efficient parallel writing. A dedicated control process dynamically manages these phases based on the status of each process. To further refine the I/O strategies, we collect performance data from the target machine to build a simulated environment. A reinforcement learning agent is deployed in this environment to identify and test better parameter configurations. Experiments conducted on two models, GOMO1.0 and LICOM3, show that this method increases output efficiency by factors of 1.54 and 13.1, respectively, compared to the commonly used PnetCDF and MPI-IO. These results suggest that this approach can significantly reduce the overhead associated with data output, providing a promising solution for enhancing the performance of climate models.

## 1 Introduction

Earth system models are essential for understanding past climate and environmental evolution and predicting future global change scenarios. Over recent decades, these models have seen significant advancements, primarily through temporal and spatial resolution improvements. However, this progress has led to exponential increases in computational demands and data output volumes. While supercomputer processing capabilities continue to advance rapidly, following Moore's Law and ensuring computational speed for high-resolution models, I/O speeds have not kept pace. This disparity has resulted in a growing imbalance between computation and data handling capabilities, particularly in high-performance computing environments. Consequently, I/O performance has emerged as a significant bottleneck in model execution. The effects of this bottleneck are particularly significant in high-resolution models, where experiments can run for months and produce several hundred terabytes data. The massive data generated by these extensive simulations greatly impacts overall performance, underscoring the urgent necessity for enhanced I/O strategies in Earth system models.



To enhance I/O performance in models, Corbetty et al. (1996) proposed advanced parallel I/O systems based on the Message

Passing Interface (MPI), including MPI-IO. Similarly, PnetCDF (Li et al., 2003) was developed as a parallel I/O library that leverages MPI for efficient data handling. PIO (Dennis et al., 2012) also utilizes MPI to optimize input/output operations in high-performance computing environments. Additionally, ADIOS (Lofstead et al., 2008; Lofstead et al., 2009) incorporates MPI to provide a flexible and high-performance I/O framework suitable for various scientific applications. These systems are widely used in models and employ parallelization and optimization techniques to leverage multiple processors or nodes for

input and output operations, thereby improving data access efficiency. The overall runtime of simulations consists of two main phases: compute time and I/O time. Despite the libraries above help reduce I/O time for large-scale data, iterative simulations still require the compute phase to wait for the finish of I/O. Essentially, the I/O and compute phases remain serially dependent on each other. There is potential to improve I/O efficiency by overlapping these phases. To address this, we previously introduced CFIO1.0 (Huang et al., 2014), adding additional dedicated processes specifically for I/O tasks. As shown in Fig. 1,

When the computing process needs to output files, it sends data to the I/O-dedicated process via the MPI message interface and then proceeds to the following computation step. The actual I/O operations are handled by the I/O process, which automatically overlaps the I/O phase with the computation phase, thereby reducing the overall time of numerical simulations. In this paper, we also adopt the principles of CFIO1.0 by introducing an additional control process instead of a set of I/O processes. This control process utilizes the common issue of load imbalance among different processes in climate models,

which arises from slight variations in hardware resources such as CPUs and differences in the computational workload assigned to each process. By leveraging this imbalance, the control process directs slower computing processes to forward their output data to faster ones. The faster processes then form multiple sub-communication domains to perform collective I/O. This approach effectively transforms the disadvantage of load imbalance into an advantage by filling idle time during computations with I/O operations, thereby maximizing the utilization of hardware resources. Compared to our previous version,

CFIO1.0, we only introduce a single additional control process rather than a set of dedicated I/O processes. Furthermore, we have designed and implemented an automated parameter selection solution to ensure that users from diverse backgrounds can fully take advantage of the parallel I/O methods presented in this paper for climate models. This solution analyses I/O information collected from pre-execution on the target machine and employs reinforcement learning techniques in a simulated environment to identify optimal strategy parameters. Once these parameters are determined, they are applied during model

execution to enhance the efficiency of parallel I/O operations. The above work has resulted in the second version, CFIO 2.0. We tested the methods presented in this paper using two climate models: the Regional Ocean Model (GOMO 1.0, Huang et al., 2020) and the ocean general circulation model (LICOM, Yu et al., 2012). GOMO achieved a speedup of 2.5 times at a resolution of 5 km using 4800 CPU cores, while LICOM attained a speedup of 13.1 times at a resolution of 10 km with 19200 CPU cores compared to their original performance. Additionally, we compared our method's performance against PnetCDF

and PIO, finding that our approach demonstrated superior performance in large-scale settings. The rest of the paper is organized as follows: Chapter 2 outlines the motivations behind our approach. Section 3 details the design and architecture of the proposed solution. Section 4 offers an overview along with a straightforward example of its interface. Section 5 presents a





performance evaluation and analysis. Section 6 reviews related work in the field. Finally, conclusions and potential areas for improvement are discussed in Section 7.

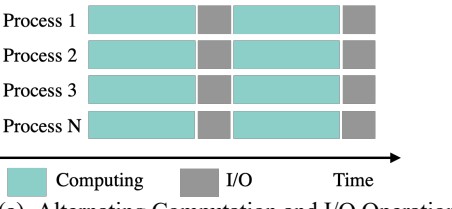


(a). Alternating Computation and I/O Operations

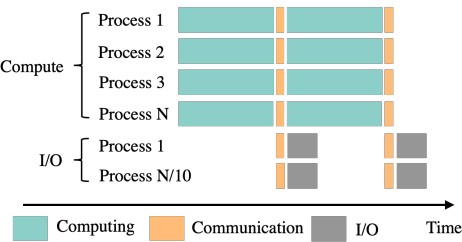

(b). Parallel Computation and I/O Operations

**Figure 1: Concurrent I/O and Computation.**

## 2 Motivation

### 2.1 Utilizing I/O aggregation and I/O forwarding

Most parallel I/O methods use the ideas of I/O aggregation and I/O forwarding. I/O aggregation refers to the simultaneous execution of I/O operations by a group of processes, where data is read or written in a coordinated manner. This method

aggregates multiple requests into fewer, more extensive I/O operations, significantly improving throughput and reducing the overhead associated with individual I/O calls. In contrast, I/O forwarding means directing I/O requests from multiple computational processes to fewer dedicated I/O processes, thereby minimizing communication overhead and reducing concurrent access to the file system. This approach enhances resource utilization and reduces idle time. By leveraging both I/O aggregation and I/O forwarding, supercomputing systems can achieve better performance and scalability, effectively

managing the challenges posed by large-scale data transfers. As shown in Fig. 2, PnetCDF employs I/O aggregation across all processes for its output logic. In the case of PIO, data is first transmitted to a small subset of processes using I/O forwarding, and then I/O aggregation is used to write the data to the parallel file system. In summary, by employing I/O aggregation and I/O forwarding, we can transform inefficient global parallel I/O requests into a mechanism where data is transmitted over the network to a small subset of processes, which then utilize I/O aggregation techniques to perform the actual I/O operations.





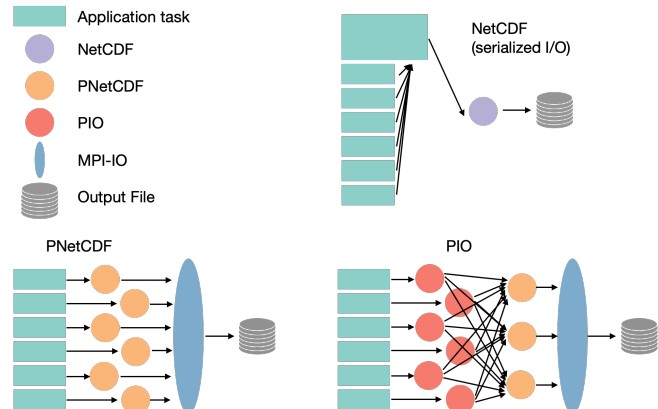

**Figure 2: I/O aggregation and I/O forwarding in PnetCDF and PIO.**

## 2.2 Dynamic scheduling in load imbalance scenarios

Load imbalance among processes is a prevalent challenge in climate models. For example, in ocean models, global or regional grids often contain numerous land grid points that do not contribute to ocean dynamics. This results in processes managing more ocean grid points facing a heavier computational burden, while those with more land grid points experience lighter workloads. Similarly, atmospheric models employ parameterization schemes to account for small-scale processes that influence large-scale flow fields, leading to varying usage frequencies based on regional meteorological conditions. Despite algorithms designed to distribute tasks evenly, performance disparities persist in supercomputing environments due to variations in CPU performance, network communication latency, bandwidth limitations, and fluctuations from operating system scheduling. Consequently, the execution speed is often constrained by the slowest process, which others must wait for at synchronization points. To optimize I/O operations, we propose leveraging these waiting periods by allowing faster processes to handle I/O tasks while slower ones complete their computations.

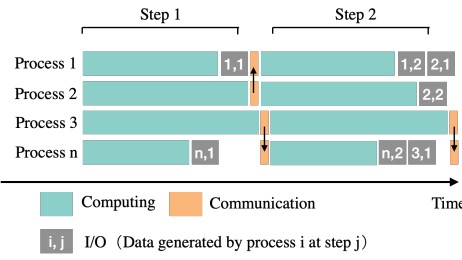

**Figure 3: Exploit the load imbalance and insert I/O into the waiting gap.** (After the conclusion of the first computational step, a dynamic data handling process is initiated. Process 1, having completed its calculations, stores its data in a buffer and then receives data from Process 2. Due to its slower processing speed, process 2 transfers its data to Process 1 via MPI before moving to the next computational step. Similarly, Process 3 forwards its data to Process n. Process n, after storing its data in a buffer, receives the transferred data from Process 3. In the second step, Process 1 caches its newly generated data and the data received from Process 2 in the previous step. Process 2 only caches its current step's data. Process 3 continues to transfer its data to Process n. Process n caches its new and previously received data from Process 3.)



Figure 3 illustrates how data is forwarded from faster processes to slower ones, leveraging the differences in computational speed among the processes. This strategy enhances I/O efficiency by utilizing idle processing time and helps balance the workload, thereby mitigating the effects of computational imbalances. Integrating I/O operations during these pauses can improve overall system performance without requiring the additional hardware resources introduced in CFIO1.0.

## 2.3 Selecting the appropriate I/O pattern

There are five fundamental I/O approaches on a large scale, outlined in Table 1. The first approach is the serial output mode of NetCDF, which has a long output time but does not require post-processing. The second method utilizes PnetCDF's parallel I/O, which provides fast output speeds for a small to moderate number of processes. However, as the number of processes increases to thousands or even tens of thousands, the output speed does not scale linearly due to the requirement for global communication operations, which become increasingly time-consuming with larger process counts. The third method allows each process or thread to output data to a separate file, which can then be merged into a complete NetCDF file through post-processing. While this approach often results in faster output times, having too many files can slow down the distributed file system's performance when handling metadata, leading to variable output time that may be short or moderate. In the fourth scenario illustrated in Fig. 4, black numbers represent process IDs, and yellow areas denote I/O domains. Each domain outputs data corresponding to a specific region, dividing a temperature field into six parts across six communication domains that operate independently. One variable is divided into six files, necessitating post-processing to concatenate them into a complete NetCDF file. Lastly, in the fifth approach, different physical quantities are forwarded to distinct domains; for example, domain 1 receives global temperature data and performs parallel output within its communication domain, while domain 2 handles global pressure data similarly. This allows different domains to process different data at the same time. The final pattern offers a shorter output time and does not require post-processing, making it the optimal choice for large-scale scenarios.

**Table 1 Five fundamental I/O approaches on a large scale.**

| Write pattern | Number of output files | Run time | Postprocessing time |
|---|---|---|---|
| Single-threaded, single-file | 1 | Long | None |
| Parallel I/O, single shared file | 1 | Moderate | None |
| Distributed I/O, single file per PE | PEs | Short or Moderate | Long |
| Single file per I/O domain, single area per domain | I/O domains | Short | Long |
| Single file per I/O domain, single variable per domain | I/O domains | Short | None |





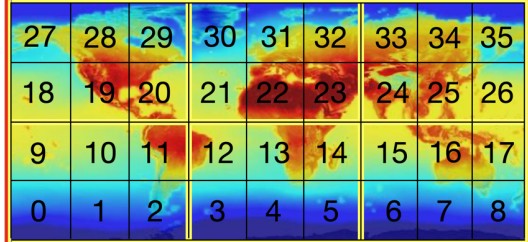

**Figure 4: A representative division of a global domain is illustrated, where the black squares denote the computational areas assigned to each process. The yellow lines indicate how these computational areas are combined to form a larger I/O domain, while the red line delineates the boundaries of the entire global domain.**

## 3 Design and Implementation

This chapter comprehensively describes the proposed method's design and implementation. We begin with a detailed explanation of the system architecture, which divides the I/O workflow into three phases: dynamic forwarding, data rearrangement, and data output. Each of these phases will be discussed in detail. Finally, we address the various trade-offs encountered during the design and implementation phases and the selection of parameters by utilizing a reinforcement learning-based simulation environment to determine the optimal operational parameters.

### 3.1 System architecture

We divide the entire I/O workflow into three phases: data forwarding, data rearrangement, and data output:

**Data forwarding.** The purpose of data forwarding is transfering the output data from each process to a limited number of other processes via the MPI communication interface over the network. This approach prevents all processes from simultaneously initiating I/O operations on the disk, a common practice in parallel I/O. For instance, in the PIO framework, data is forwarded to a specific subset of processes. In CFIO 1.0, data is forwarded to an additional group of processes designated for I/O tasks. In the method proposed in this paper, data is forwarded to processes that have faster computation speeds and have completed their tasks earlier. Further details will be discussed in 3.2.

**Data rearrangement.** Data rearrangement aims to reorganize the data received during the data forwarding phase. For instance, if there are 100 processes in the data forwarding phase and 4 of these processes are selected to receive the global temperature field data, each of these 4 processes will hold a portion of the temperature data. However, the data chunks they each possess may be disorganized. To address this, the 4 processes must exchange data to rearrange it, aiming to make each process's 25 data blocks contiguous in memory. This way, each process can effectively form a larger, more coherent data block. Further details will be discussed in 3.3.

**Data output.** This is the phase in which data is written to disk. In the previous two phases, the data was temporarily stored in the buffers of specific processes but had not yet been written to disk. This phase involves two main components: the construction of sub-communication domains and the data output itself. For example, the temperature field data was forwarded



and temporarily stored in the buffers of processes 1, 2, 5, and 6 during the earlier phases. In contrast, the pressure field data was stored in the buffers of processes 3, 4, 8, and 9. The first step is to create two sub-communication domains: {1, 2, 5, 6}

and {3, 4, 8, 9}. Each domain will then use PnetCDF's collective I/O mode to write the data to disk. Further details will be discussed in 3.4.

The system architecture is illustrated in Fig. 5. At the center is an additional control process that coordinates all tasks and records global information. Specifically, it manages forwarding controls, communication domain controls, data output controls, and data logging. Surrounding this control process are the ordinary processes that perform integration calculations while

running models. When these processes need to output data, they send a request to the control center and receive commands to execute specific actions. In the upper right corner of Fig. 5, two processes form a communication domain, within which PnetCDF collective I/O is invoked to write data to the file system.

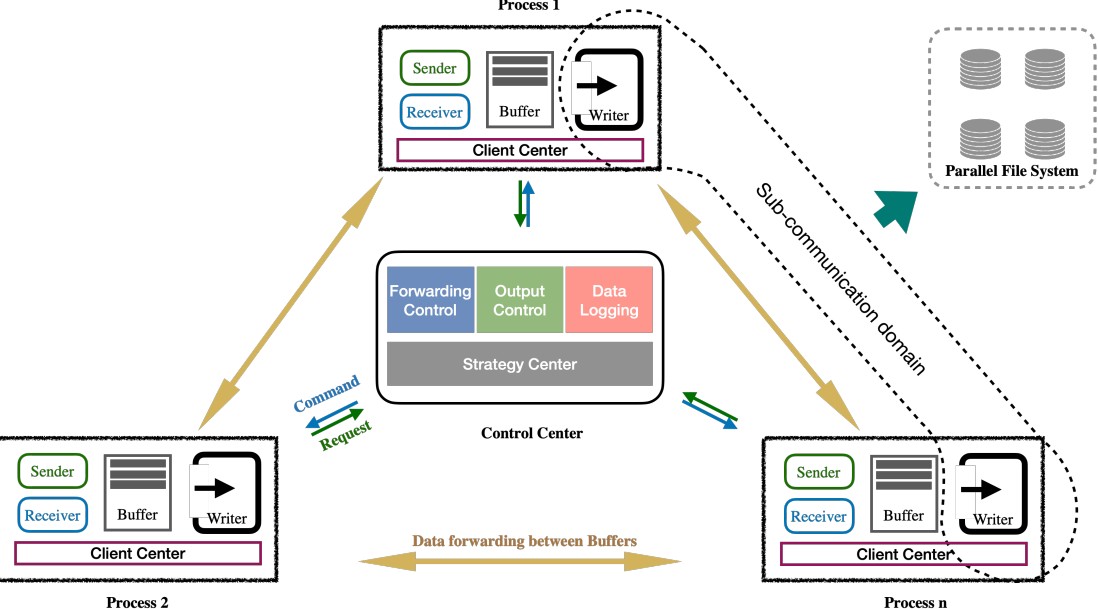

**Figure 5: The system architecture.**

**3.2 Phase 1: Data forwarding**

This section explains the data forwarding mechanism employed in our proposed method. Data forwarding is a crucial intermediate step in the I/O workflow, facilitating the efficient transfer of data from multiple source processes to a selected group of target processes. Utilizing the MPI communication interface ensures that I/O operations are controlled, thereby minimizing contention for disk access.

The core issue of data forwarding is determining "who sends to whom". We adhere to the following principles: First, the receiving processes should ideally compute quickly and complete their tasks early. This allows slower processes to move on to the next round of computations after sending their data, thereby balancing the workload among processes and enabling faster



processes to contribute effectively rather than remaining idle, as in traditional I/O scenarios. Second, the receiving processes should be better distributed across different nodes. In modern CPU clusters, processes on the same node share the network
bandwidth. If multiple processes on the same node simultaneously perform MPI data reception tasks, it can slow down data transfer rates. Third, before data is written to the file system, a process should be responsible for caching only one variable at a time. For example, if Process 2, which computes quickly, is selected to receive the variable *"temperature"*, it will not be chosen again for data reception of *"pressure"* before data output, even if its speed remains high. This policy prevents any single process from being overloaded with data, which could strain node memory and complicate the subsequent data
rearrangement and output phases due to increased resource competition.

Figure 6 illustrates the flowchart for each process within the data forwarding phase. The red section represents the additional control process we have introduced, which manages the tasks executed by each process. The blue portions indicate the beginning and end of the data forwarding phase. During these phases, all processes have equal roles and later become I/O processes or regular computation processes based on the control process's decisions. The yellow sections denote the I/O
processes chosen for their high computation speed. These processes serve as buffers and are responsible for receiving data from other processes under the control process's direction, temporarily storing it in memory. The green sections represent the regular computation processes that immediately proceed to the following computation phase after sending data to the I/O processes. After the data forwarding phase, the output data will be temporarily stored in the memory of specific processes.

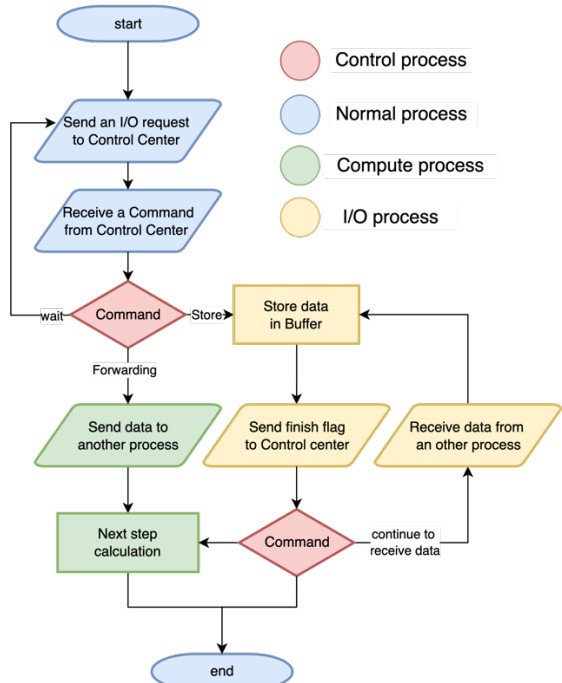


**Figure 6: Flowchart of each process within the forwarding phase.**



The control process continuously receives messages from other processes to monitor their states—whether they are finalizing, requesting write operations, or needing I/O services. When a process requests I/O, the algorithm evaluates its ability to handle the task based on its current load and resource availability. If the process is deemed eligible, it is assigned as the I/O process;

otherwise, the task is delegated to another process. Finally, the algorithm manages the completion of I/O tasks by updating states and verifying whether all tasks are finished, ensuring that all processes can continue their computations without interruption. This structured approach enhances data management efficiency in parallel I/O operations and minimizes process contention.

### 3.3 Phase 2: Data rearrangement

This section discusses the data rearrangement process, essential for organizing the data received during the forwarding phase. After the data forwarding phase, the selected processes may hold fragmented data blocks that must be organized for efficient access and processing.

Figure 7 illustrates the data rearrangement process. The black boxes in the figure represent processes, with eight processes involved in this example. During Phase 1, data is sent to Processes 1 and 6 via data forwarding. If these processes were to

directly write the data to the file system at this stage, it would be inefficient because both contain multiple fragmented data segments that are not contiguous. Therefore, we employ data rearrangement to reorganize the data within Processes 1 and 6. After this rearrangement, as shown in the lower part of Fig. 7, each process will hold a single large data block composed of four contiguous data segments. This allows Processes 1 and 6 to perform a more efficient write operation to the disk, as they will each write a more big, contiguous data block.

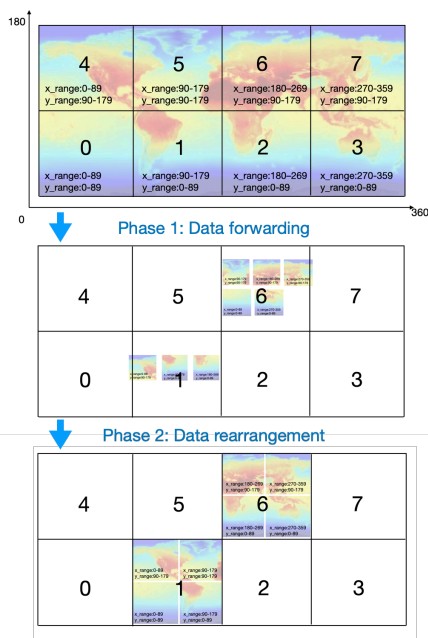


**Figure 7: Data rearrangement.**



The Data Exchange and Rearrangement Algorithm, as shown in Algorithm 1, utilizes MPI to exchange data among multiple processes. Since only a few processes contain data buffers, meaning that only a few processes on the same compute node receive the data, OpenMP multithreading is employed to rearrange the received data into a continuous block. Initially, the
algorithm creates a custom MPI data type to define the structure of the data being exchanged. It then gathers metadata from all participating processes to understand the distribution of this data. Next, the algorithm integrates and sorts this metadata based on the starting positions of the data blocks, preparing for data exchange by mapping each block to its target process. The algorithm's core involves sending and receiving data using non-blocking MPI operations, which allows processes to continue their computations while waiting for communications to complete. Finally, it ensures that all ongoing communications are
completed, allowing all processes to have their necessary data arranged into one big continuous block efficiently for further processing.

**Algorithm 1: Data exchange and rearrangement algorithm.**

---
**Algorithm 1** Data Exchange and Rearrangement Algorithm
---
1: // Step 1: Create custom MPI data type
2: Create meta data with MPI data type $mpi\_piece\_of\_data$ containing fields of $start/count/location/address$ within each process
3: // Step 2: Collect meta data
4: Use $MPI\_Allgather$ to collect all the metadatas from each process into $all\_meta$
5: // Step 3: Sort meta data
6: Sort $all\_meta$ based on the $start$ and $count$
7: // Step 4: Prepare for data exchange
8: **for** each data block in local metadata **do**
9:     **if** the target process is not equal to the local rank **then** Remove this block from the metadata
10:     **end if**
11: **end for**
12: // Step 5: Send and receive data
13: **for** each integrated metadata block **do**
14:     **if** the target process is equal to the local rank and the current location is not equal to the local rank **then** Use $MPI\_Irecv$ to receive data
15:     **end if**
16:     **if** the current location is equal to the local rank and the target process is not equal to the local rank **then** Use $MPI\_Isend$ to send data
17:     **end if**
18: **end for**
19: // Step 6: Wait for all communications to complete Use $MPI\_Waitall$ to wait for all non-blocking requests to complete
20: // Step 7: Concatenate multiple data blocks into a contiguous data block using OpenMP within each process
---

## 3.4 Phase 3: Data output

This section will delve into the specifics of utilizing PnetCDF's collective I/O mode for efficient data output to disk. Collective I/O is a method that allows multiple processes to coordinate their I/O operations, significantly improving performance and reducing the overall time taken to write large datasets.

Before outputting the data, defining and establishing communication domains is necessary. As illustrated in Fig. 8, after several rounds of forwarding, the data to be output is stored in different processes. Under the coordination of the control process, those
processes that cache the same variable are grouped into a sub-communication domain.





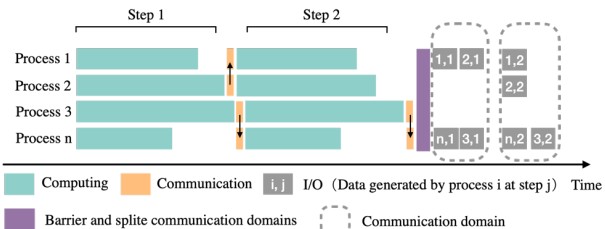

**Figure 8: Create new communication domains.**

Next, we will utilize PnetCDF's collective I/O mode to write the data. Collective I/O enhances data writing efficiency by reducing I/O bottlenecks and improving throughput. Allowing multiple processes to coordinate their write operations

minimizes contention for disk access and consolidates write requests into fewer, more significant transactions. This approach not only reduces latency but also ensures data consistency across processes. Additionally, collective I/O scales well with larger datasets and a higher number of processes, making it an effective solution for high-performance computing environments.

We note that different data have been cached in separate processes during the initial forwarding and rearrangement phases. At this stage, the data have been organized according to communication domains, ensuring that cached processes for the same

variable reside within the same domain. Consequently, different communication domains can simultaneously utilize PnetCDF for output. For example, as shown in Fig. 9, the *"temperature"* is located in MPI domain 1, while *"pressure"* is in MPI domain 2, allowing for simultaneous output. This approach effectively maximizes the bandwidth of the file system.

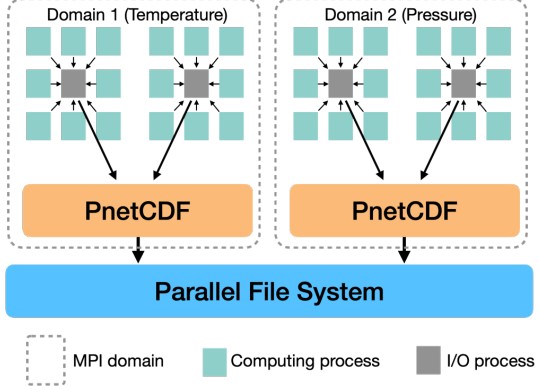

**Figure 9: Output with PnetCDF.**

**3.5 Select better parameters with reinforcement learning**

As previously discussed, the Control center manages all scheduling processes. These strategies involve several trade-offs that must be considered to maximize performance. The following points highlight some of the most essential trade-offs:

**Trade-off 1**, the timing of file output operations, presents a trade-off in our system design. As shown in Fig. 8, we perform an output operation after two cycles of calculation and I/O forwarding, similar to MPI asynchronous communication with

*MPI_Isend* and *Wait*. Data is forwarded to some processes when a user launches an I/O request. However, deciding when to write this data to the distributed file system is complex. Writing too early does not fully leverage the advantages of



simultaneous output from multiple communication domains. Conversely, delaying the write operation risks saturating process buffers, complicating forwarding, and potentially causing buffer overflow errors. This balance between minimizing communication overhead and optimizing buffer utilization requires careful consideration.

**Trade-off 2**: The number of I/O processes presents a significant trade-off in our system design. For instance, while the variable *"temperature"* could be stored in the buffers of processes 1, 4, 7, and 9, it could be stored in fewer processes, such as 1 and 4. Utilizing more processes for storage offers the advantage of significantly reducing communication time during the forwarding phase. In a system with $n$ processes, where $m$ processes are used for data storage, each process receives data blocks from an average of $n/m$ processes. Despite employing asynchronous MPI communication, data transmission remains serial at the lowest

level; thus, a more considerable $m$ value decreases forwarding time. However, during the output phase, when the number of processes in a sub-communication domain becomes too large, the utilization efficiency of the parallel file system does not scale linearly. This balance between communication efficiency and file system performance must be carefully optimized based on specific system configurations and computational requirements.

**Trade-off 3**, the selection of I/O processes presents a trade-off in our system design. Consider the scenario where we have

decided to use 4 processes as the I/O processes for variable *"temperature"* A straightforward approach would be to designate the earliest four processes that launch the I/O request for *"temperature"* as I/O processes. However, this could lead to performance issues due to the nature of high-performance computing architectures. In climate models, the exchange of Halo information often results in adjacent processes having similar computation speeds. Logically adjacent processes are often located on physically adjacent CPU cores within the same compute node. This can lead to network congestion when many

send data to adjacent processes, sharing the same node's network interface and bandwidth. Therefore, deciding between selecting the earliest 4 available processes or waiting longer to select processes from different nodes represents a trade-off.

While identifying an optimal strategy is challenging, it is essential for climate model simulations that run for extended periods on stable systems. To address this issue, we draw on techniques from the field of automation that utilize reinforcement learning. We create a virtual environment by collecting data to identify the target machine's necessary parameters. A reinforcement

learning (RL) agent is employed in this virtual setting to discover better parameter configurations. Once optimal configurations are identified in the virtual environment, they can be applied to accurate model runs, maximizing I/O efficiency.

RL is a branch of machine learning that focuses on how intelligent agents should act in an environment to maximize cumulative reward. RL does not require labeled input/output pairs or explicit correction of suboptimal actions. Instead, it emphasizes learning through interaction with the environment, making it particularly suitable for complex, dynamic systems such as our

I/O optimization problem.

Figure 10 illustrates this process. First, we run a test program on a real machine to collect multiple sets of required parameters, including network communication speed, PnetCDF output rates at various scales, node memory details, and CPU information. Using these data, we construct a virtual environment. Based on this virtual environment, we employ a reinforcement learning agent to perform simulated I/O tasks, identifying the optimal strategy for that environment. Finally, we apply the best strategy

to the natural machine environment for I/O tasks.





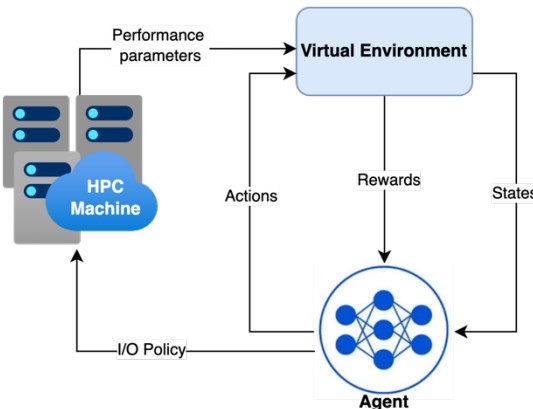

**Figure 10: Select better parameters with reinforcement learning.**

Table 2 presents critical configurations for constructing the reinforcement learning environment. First, we execute a test program in a natural environment to collect performance data from the machine, including the throughput of PnetCDF, the

speed of MPI communications and hardware information. This data is then used to create a virtual reinforcement learning environment that simulates the I/O process during model running. When a new variable requires I/O, the agent must take an action. There are three possible actions: the first is to continue forwarding the variable or execute Phase 3, which writes the previously cached variables to the file system. This corresponds to the trade-off mentioned earlier (Trade-off 1). The second action involves determining the number of I/O processes to cache the variable, relating to the previously described trade-off

(Trade-off 2). The third action addresses a scenario where an idle process is available for I/O, but it resides on the same node as another I/O process. As discussed in Trade-off 3, waiting for a different idle process on a separate node may be more beneficial. Thus, the third action specifies the maximum wait time allowed. If this wait time is exceeded, we will designate the idle process on the same node as an I/O process to avoid excessive delays. Regarding the states, we define them based on the number of currently cached variables and the number of I/O processes in use. After executing Phase 3, these values are reset

to zero. For the reward mechanism, we use the total execution time in the simulated environment as the reward.

**Table 2 Key configurations in reinforcement learning methods.**

| Collected data | Actions | States | Reward |
|---|---|---|---|
| 1. Throughput of PnetCDF<br>2. Speed of MPI<br>3. CPU, Memory | 1. Execute Phase 1 or 3<br>2. Number of I/O processes<br>3. Number of waiting rounds | 1. Number of cached variables<br>2. Number of total I/O processes | Total running time |

## 3.6 Improvements from CFIO1.0 to CFIO2.0

First, the number of additional I/O processes required has been significantly reduced. As described above, Fig. 1(b) illustrates

the approach of CFIO1.0, which necessitates a batch of extra processes dedicated solely to I/O. For instance, if the model itself





uses 1,000 processes for computation, an additional 200 processes are required specifically for I/O. In contrast, with version 2.0, we only need to add one extra control process. Thus, for a model utilizing 1,000 processes, the number of additional processes required has been reduced from 200 to just 1.

Second, in the I/O output phase, CFIO 1.0 relies on a fixed number of processes to handle I/O operations. However, this fixed
count may not be adequate to fully exploit the parallel capabilities of the file system. Increasing the number of processes could lead to unnecessary waste of hardware resources. Conversely, version 2.0 theoretically allows the number of I/O processes to match that of the computing processes since I/O processes are dynamically selected from the pool of computing processes. This flexibility in selection can significantly enhance the parallel performance of the file system. The theoretical maximum output speed is faster than version 1.0

In version 1.0, a significant bottleneck arose from the MPI system's allocation mechanism. For instance, when 200 processes were designated as I/O processes, they were concentrated on just four nodes, with 64 cores on each node. This limitation meant that only the bandwidth of these four nodes could be utilized for data forwarding. However, version 2.0 addresses this issue by introducing dynamic scheduling to enhance forwarding speed. In this new version, processes that complete their computations quickly are designated as data receivers. Furthermore, the system aims to distribute these receivers across
different nodes whenever possible, thereby maximizing the utilization of network bandwidth across the nodes.

## 4 The CFIO2 interface

Since the NetCDF format is widely recognized as the standard data format in the climate community, we have decided to adopt it. This choice aims to lessen the workload involved in updating code and processing data during the transition to CFIO 2.0. Creating a new dataset in NetCDF requires several essential steps: initializing the dataset, defining dimensions, variables, and
attributes, entering data mode, writing the variable data, and closing the dataset file. CFIO 2.0 simplified this procedure, as shown by the interface functions listed in Table 3. Additionally, this interface supports programming in both C/C++ and Fortran.

To promote consistency across all computational processes, all functions within CFIO 2.0 are designed as collective I/O operations. For instance, when a climate model generates a new dataset, it is crucial that all computing processes execute the
CFIO 2.0 functions in the same order and with matching arguments.

**Table 3 Main Interfaces of CFIO2.**

| Function | Description |
|---|---|
| cfio2_init( | Initialization of CFIO2 |
| comm, | Communication domain |
| IO_process_per_node, | Maximum number of I/O processes allowed on each node |
| IO_process_per_var) | The maximum number of processes allowed per variable |



| cfio2_put_vara( | Finalization of CFIO2 |
| --- | --- |
| all_comm, | Communication domain |
| filename_in, | File name |
| var_name, | Variable name |
| datatype, | Data type |
| dim_name | Names for each dimension |
| global, | Global varialble size |
| start, | Starting position of the current process in the global array |
| count, | Size of the data in current process |
| buf, | Buffer of the data |
| append) | Append or overwrite to nc file |
| cfio2_wait_output( | Wait for data to actually be written to disk |
| comm) | Communication domain |
| cfio2_finalize( | Finalization of CFIO2 |
| comm) | Communication domain |

Listing 1 provides a simple illustration of how to use CFIO 2.0 for data output. The cfio2_init function is used for initialization and accepts two parameters: IO_process_per_node and IO_process_per_var. These parameters specify the maximum number of I/O processes allowed on each computing node and the maximum number of I/O processes that can be associated with a single variable, respectively. The cfio2_put_var function outputs variable data, while the cfio2_wait_output function operates similarly to MPI_Wait, serving as a wait statement until the actual output of the variable is complete.

**Listing 1 A simple example with CFIO2.**

```
cfio2_init(MPI_COMM_WORLD, IO_process_per_node, IO_process_per_var);
if (not Crontrol Process)
{
        //Compute
        cfio2_put_vara(MPI_COMM_WORLD, filename_1, varname_1,
                        cfio2_data_type_double, dim_name_1,
                        global_1, start_1, count_1, data_1, append);
        //Compute

        cfio2_put_vara(MPI_COMM_WORLD, filename_1, varname_1,
                        cfio2_data_type_double, dim_name_2,
                        global_2, start_2, count_2, data_2, append);

        cfio2_wait_output(MPI_COMM_WORLD);
}
cfio2_finalize(MPI_COMM_WORLD);
```

## 5 Experiments

The experiments were conducted on the supercomputer of the Earth System Numerical Simulation Facility, "EarthLab". This supercomputer features Hygon C86 7185 processors, with each node equipped with 64 cores and 256GB of memory. The nodes are interconnected via the InfiniBand network and utilize the Dawning ParaStor parallel file system. The operating





system is CentOS Linux release 7.6, and we used the Intel compiler version 17.0.5. The MPI environment is based on the optimized HPCX v2.7.4, which extends Open MPI.

In the following sections, we evaluate CFIO2 using a regional model, GOMO1.0, and a global model, LICOM3.0, comparing its output performance with PnetCDF and PIO. We chose PnetCDF and PIO because they are widely used parallel I/O libraries in Earth system models and offer relatively good performance. Next, we conduct a more detailed comparison and analysis of

throughput across various scenarios. Finally, we analyze CFIO2's speedup.

## 5.1 GOMO case study

The Generalized Operator Model of the Ocean (GOMO1.0) is a ocean model developed from the Princeton Ocean Model (POM). By utilizing OpenArray (Huang et al., 2019) to abstract parallel computing details, GOMO is streamlined to just 1860 lines of Fortran code, enhancing its clarity and ease of maintenance. Moreover, GOMO exhibits robust scalability and

portability across multiple platforms, including CPUs and the Sunway architecture (Fu et al., 2016). GOMO utilizes a bottom-following free-surface staggered Arakawa C grid and implements a mode-splitting algorithm. Adapted from POM, the model's core equations are transformed into operator expressions that OpenArray processes efficiently.

In this experiment, we employed the GOMO model to conduct a simulation with a resolution of 5 kilometers, covering the North Pacific, North Indian Ocean, and South China Sea regions (5°S to 45°N, 50°E to 90°W). The simulation duration was

set for 1 days, during which we utilized a preprocessing scheme to generate a 5 km resolution grid. The grid was structured using latitude and longitude coordinates, resulting in an east-west grid spacing of 3769 to 5560 meters and a north-south grid spacing of 5569 meters. A restart file was created for each simulated day, and the total size of the output files generated during the simulation was approximately 700 GB.

The original GOMO model managed file output using the OpenArray I/O interface, which relies on PnetCDF for parallel I/O

operations. In this study, we will implement an alternative method for file output and compare the model's total runtime under three scenarios: without file output, using the original interface for file output, and utilizing the method proposed in this paper. We tested configurations with 900, 1800, 3600, and 4800 cores for these simulations.

We recorded the overall running time of the GOMO model using the CFIO2 method and compared it with the running times achieved using the default I/O method and a no-I/O scenario. The running time in the no-I/O case represents pure computation

time, serving as the upper limit for the maximum performance attainable through the complete overlap of I/O and computation. The results are illustrated in Fig. 11. In the figure, CFIO2(default) refers to the parameter strategy selected manually for this model and machine. At the same time, CFIO2(RL) indicates the configuration strategy derived from reinforcement learning applied in a virtual environment, as described in Section 3.5.

As expected, CFIO2 outperformed the default I/O method in the GOMO model. The total running time with 900 computing

processes decreased from 1853 seconds to 1798 seconds (default) and 1753 seconds (RL), resulting in a performance improvement of 1.06 times for GOMO. The acceleration effect was more pronounced with 4800 computing processes, where the total computation time was reduced from 948 seconds to 612 seconds, achieving a speedup of 1.54 times.



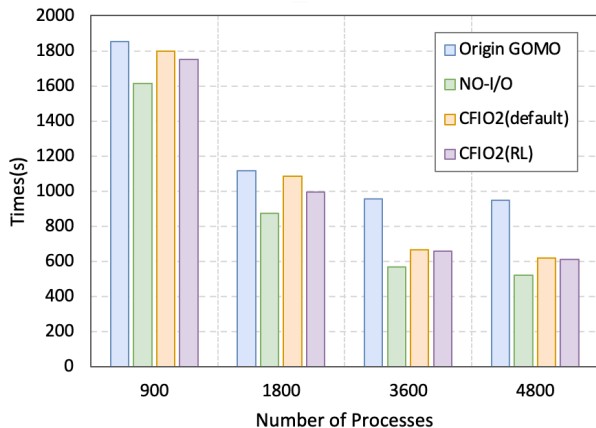

**Figure 11: The overall running time of GOMO with different I/O approaches.**

## 5.2 LICOM case study

LICOM3.0 is an ocean model which serves as the sea component of the LASG/IAP Earth System Model, known as FGOALS_g2. LICOM employs a two-dimensional data decomposition of the horizontal domain to partition data arrays evenly across all computing processes. The model uses an MPI-IO interface for output files every 4 model hours.

In this experiment, we utilized LICOM version 3 with a resolution of 0.1°, simulating 10 days. The output variables consist of three-dimensional arrays of size 3600×2302×55. The total size of the output files is about 1020 GB.

The results are illustrated in Fig. 12. The green area labeled "no-IO" represents the pure computational time of LICOM3. The time decreases from 5687 seconds with 1200 processes to 474 seconds with 19200 processes, achieving a speedup of 12 times. This indicates a parallel scalability of 75%, demonstrating solid efficiency in parallel processing.

However, when we examine the blue section, we notice that although the total running time decreases when the number of processes increases from 1200 to 4800, the I/O time increases. Furthermore, as we scale from 4800 to 19200 processes, the total computation time rises significantly, with the overall time at 19200 processes even longer than that at 2400. Thus, we conclude that MPI-IO exhibits poor scalability in the current system when the number of processes reaches thousands or more. During our tests, we also observed that executing MPI-IO generates numerous hidden temporary files in the working directory, with the number of files equal to the number of processes. These temporary files disappear once the corresponding output is completed. This phenomenon is related to the optimization policy of MPI-IO for the file systems. The behavior of MPI-IO can vary across different file systems (such as Lustre and GPFS) due to underlying implementation differences. In some file systems, simultaneous writes by multiple processes to the same file may lead to data conflicts and inconsistencies. While MPI-IO uses temporary files to address these issues, this approach can also reduce efficiency. The ParaStor parallel file system used in our tests is less common than systems like Lustre or GPFS. Consequently, MPI-IO adopts a conservative strategy with such file systems, contributing to its poor scalability.



After implementing our method, the file output time significantly improved. At 1200 cores, the program's total runtime decreased from 7562 seconds to 5932 seconds (default) and 5903 seconds (RL). At 19200 cores, the total runtime dropped from 7015 seconds to 568 seconds (default) and 533 seconds (RL), showing a 13.14 times acceleration. Additionally, the results indicate that the strategy obtained through reinforcement learning outperforms the parameters set manually.

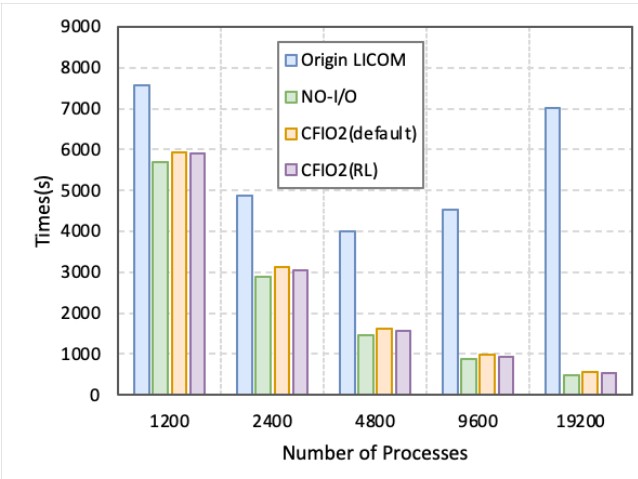


**Figure 12: The overall running time of LICOM3 with different I/O approaches.**

**5.3 In comparison to PIO and PnetCDF**

PnetCDF was developed to enable parallel I/O for NetCDF and is based on MPI-IO to leverage collective I/O optimizations.
PIO, an application-level parallel I/O library designed for the Community Earth System Model (CESM), supports multiple back-end I/O libraries. In our experiments, we utilized PnetCDF version 1.12.3 and PIO version 2.4.2, selecting PIO_IOTYPE_NETCDF4P and PIO_IOTYPE_PNETCDF as the back-end method for PIO to achieve optimal parallel throughput.

To compare performance differences, we conducted weak scalability tests. Each process was assigned a fixed data block size
of 50x60x70 floats, considered a moderate size for single-process data blocks in climate models. Each test program was then run to output 20 variables continuously. We calculated the throughput based on the output time and the generated file size. To prevent network or file system congestion caused by other jobs, we conducted five runs for each case and selected the shortest execution time.

The results of PIO and PnetCDF are presented in Fig. 13. In the throughput testing of PIO, an important parameter is the
"stride". This parameter determines how many processes are selected to execute I/O operations, with other processes sending data over the network to the designated I/O process. This approach helps reduce bottlenecks caused by synchronous access to the file system, embodying the concept of I/O forwarding. For example, when the stride is set to 20, processes 0, 20, 40, and 60 are designated as I/O processes. In our testing setup, each node is configured with 60 CPU cores (although there are 64



physical cores, a few are reserved for system use according to machine recommendations to maximize MPI program

performance). The maximum stride setting is 60, meaning each node has only one I/O process. If the stride exceeds 60, some nodes will lack an I/O process, leading to resource waste and underutilization of network and I/O bandwidth. When the stride was set to 2 (resulting in 30 I/O processes per node), we achieved a maximum throughput of 1.88 GB/s with 512 processes. With a stride of 20 (3 I/O processes per node), the maximum throughput reached 3.15 GB/s at 2048 processes. When the stride was set to 60 (1 I/O process per node), we achieved a maximum throughput of 3.44 GB/s with a total of 6144 processes. In

comparison, for PnetCDF, a maximum throughput of 4.22 GB/s was reached with 8192 processes before it began to decline. Additionally, we added error bars to the PnetCDF line graph. Since the line represents the best throughput, the upper limit of the error bars corresponds to the line itself, while the lower limit reflects the lowest throughput recorded during the runs. This visualization clearly shows that as the number of cores increases, the throughput of PnetCDF exhibits significant fluctuations. Under the same configuration parameters, the minimum throughput is only half the maximum when run multiple times.

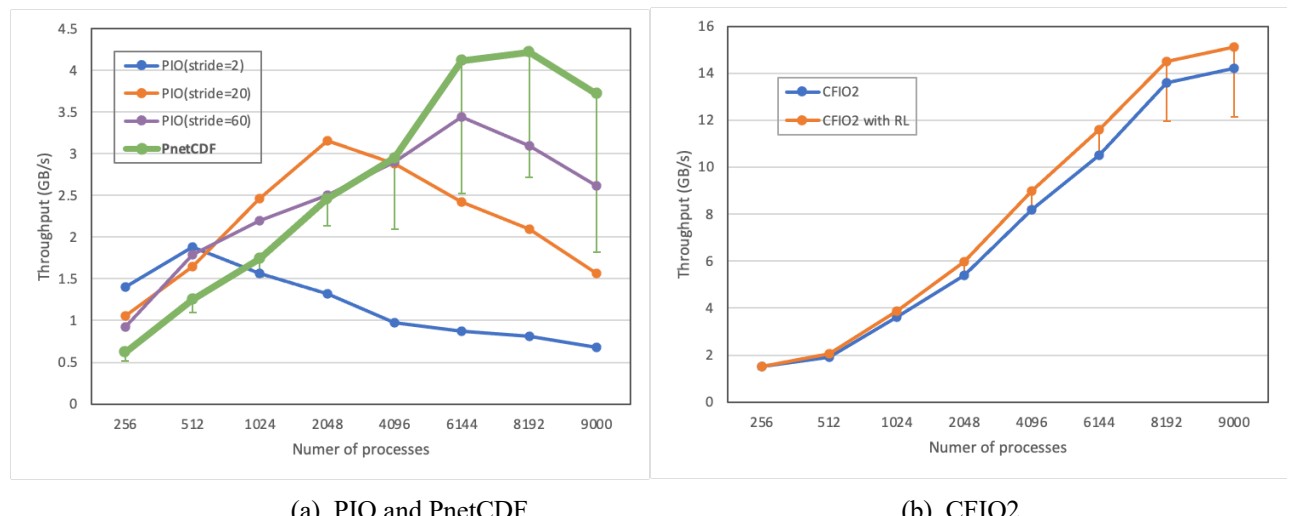


|                     |                |
|:-------------------:|:--------------:|
| (a). PIO and PnetCDF | (b). CFIO2 |

**Figure 13: Throughput of PIO, PnetCDF and CFIO2.**

On the other hand, our approach demonstrates significantly better performance. As shown in Fig. 14, at 9000 cores, the version using default manually configured parameters achieved a maximum throughput of 14.2 GB/s. The parameter strategy

optimized through reinforcement learning reached a maximum throughput of 15.1 GB/s. This performance is still 3.57 times faster than the best throughput observed in PnetCDF and PIO methods. We added error bars to the CFIO2 (with reinforcement learning) data. The error bars indicate that its performance is more stable than that of PnetCDF. CFIO2 achieved over 80% of the maximum throughput, even in the slowest run. This stability can be attributed to our method, which employs multiple sub-communication domains that independently output individual variables during the data output phase. Each sub-communication

domain consists of fewer processes, contributing to a more stable output speed.





## 5.4 Speedup analysis

In this section, we will analyze the time expenditures of various phases of our method through experimental analysis and identify the sources of the final speedup achieved. To facilitate this, we developed a Jacobi iteration program that performs I/O operations at intervals during the iterations. We utilized 4800 processes, each handling a data block of size $50 \times 60 \times 70$
in float type, resulting in each output variable being 3.75 GB in size.

We conducted several comparative experiments, as shown in Table 4. Experiment 1 features a Jacobi iteration without any I/O operations. Experiment 2 involves 10 instances of I/O without any Jacobi calculations. Experiment 3 includes Jacobi iterations with 10 instances of I/O using CFIO2. Experiment 4 changed the Phase 3 output data to serial, meaning that multiple sub-communication domains execute the output data sequentially rather than simultaneously. Experiment 5 consists of Jacobi
iterations with 10 instances of I/O but excludes the output from Phase 3. Finally, Experiment 6 includes Jacobi iterations with 10 instances of I/O while omitting the data rearrangement from Phase 2 and the output from Phase 3.

**Table 4 Comparison of Jacobi Iteration with different I/O settings.**

| No. | Description | Running time (seconds) |
|---|---|---|
| 1 | Jacobi iteration w/o I/O | 14.5 |
| 2 | Output 10 times w/o Jacobi iteration | 4.1 |
| 3 | Jacobi iteration w/ output result 10 times | 16.2 |
| 4 | Jacobi iteration w/ output result 10 times, but serial in phase 3 | 31.3 |
| 5 | Jacobi iteration w/ output result 10 times (remove phase 3) | 14.8 |
| 6 | Jacobi iteration w/ output result 10 times (remove phase 2 and phase 3) | 14.6 |

Experiment 1 shows a pure computation time of 14.5 seconds, while Experiment 2 demonstrates an I/O time of 4.1 seconds.
Experiment 3, which alternates between computation and I/O, has a total time of 16.2 seconds, less than the combined time of Experiments 1 and 2 (18.6 seconds). This indicates that some I/O time is effectively hidden during the waiting periods of computation. In contrast, Experiment 4 exhibits a significantly longer duration than Experiment 3 because it employs serial output data. In this setup, Communication Domain 1 first calls PnetCDF to complete the output of Variable 1, followed by Communication Domain 2, which outputs Variable 2. When comparing Experiments 5 and 6, we find that the time taken for
Phase 2, the data rearrangement stage, is 0.2 seconds. Additionally, comparing Experiment 6 with Experiment 1 reveals that even with data forwarding, execution time only increases by 0.1 seconds. This suggests that data forwarding is also hidden during waiting periods.

From these results, we can draw two conclusions: first, we have successfully concealed a portion of the time within the computation waiting periods; second, utilizing sub-communication domains for output data has effectively improved the
utilization of the file system.



## 6 Related work

This paper expands upon the concept of Two-phase I/O, a strategy for implementing collective I/O operations within the MPI environment, initially proposed by Rosario et al. (1993). Two-phase I/O enhances parallel computing efficiency by enabling multiple processes to perform data read and write operations more effectively, reducing file access costs. The strategy consists
of two primary phases: the prefetch phase, in which I/O requests are redistributed to minimize the frequency of file access, and the completion phase, during which processes coordinate to execute the actual I/O operations. Two-phase I/O is widely employed in MPI-IO and has been incorporated into systems such as ROMIO, an open-source MPI-IO library developed by Thakur et al. (1999). Other advancements include a novel design by Kang et al. (2019) that mitigates communication overhead by aggregating requests within compute nodes and a study by Tsujita et al. (2014) demonstrating that multithreaded Two-
phase I/O can enhance collective write performance by up to 60% on the Lustre file system. Dickens and Logan (2008) examined high-performance MPI-IO methods on Lustre, underscoring critical optimizations for efficient parallel I/O. Yu et al. (2007) investigated the utility of Lustre file connections to improve collective I/O efficiency. Overall, two-phase I/O remains an essential parallel I/O optimization technique, with ongoing research to enhance its performance in massively parallel systems.

There has been a growing interest in applying artificial intelligence (AI) techniques to optimize I/O performance in high-performance computing (HPC) systems in recent years. For example, Xie et al. (2019) explored the application of machine learning to analyze and comprehend the write performance of large-scale parallel filesystems, demonstrating how AI can effectively identify performance bottlenecks. Similarly, Isakov et al. (2022) presented a multifaceted approach for automated I/O bottleneck detection, highlighting the potential of AI to enhance the efficiency of HPC workloads. Bez et al. (2022)
introduced Drishti, a tool designed to guide end-users through the I/O optimization process, underscoring the role of AI in making I/O performance enhancements more accessible. In another study, Paul et al. (2021) characterized machine learning I/O workloads, providing insights into the specific performance behaviors of these applications under leadership-scale conditions.

Recent advancements in parallel I/O optimization have significantly enhanced the performance of climate models, which
necessitate the efficient management of large volumes of data. A notable contribution is the Parallel I/O (PIO) library, developed for the CESM and other climate models (Dennis et al., 2011). The PIO library offers a high-level interface for reading and writing distributed arrays, leveraging underlying parallel file systems. The Adaptable Input Output System (ADIOS), as introduced by Lofstead et al. (2008), has also been employed in climate models such as the Global/Regional Assimilation and Prediction System (GRAPES) to address I/O bottlenecks (Zou et al., 2014). ADIOS provides a flexible and
high-performance I/O framework that supports various data formats and storage backends, enabling efficient data management in large-scale simulations. Additionally, the Climate Fast Input/Output (CFIO) library, specifically designed for climate models, optimizes I/O operations by leveraging asynchronous communication and dynamic scheduling (Huang et al., 2014). CFIO1.0 has demonstrated significant reductions in I/O time, enhancing the overall efficiency of climate simulations.



## 7 Conclusions

In this paper, we present the second version of our parallel I/O library for climate models (CFIO2.0), building upon the foundation of its predecessor. This iteration eliminates the additional processes dedicated to I/O work introduced in the first version, instead leveraging the inherent load imbalances in large-scale process calculations. Our approach utilizes process waiting intervals for data forwarding, employs an additional process for global dynamic scheduling, and transforms global parallel output operations into localized parallel operations for multiple variables. We have also incorporated reinforcement

learning techniques to collect and analyze runtime information in specific environments, training our system in simulated scenarios to develop optimized strategies. Experimental results demonstrate that our proposed method significantly enhances output efficiency and reduces overall model execution time. Future work will focus on improving the generalization capabilities of our policy, aiming to determine optimal strategies across diverse cluster environments swiftly. We will also adopt and test CFIO2 in more climate models. This paper employs reinforcement learning to search for better parameters for

CFIO2; however, this method is still experimental and requires users to collect data on specific platforms before conducting a simulation, which remains quite complex. We hope to achieve automated runtime tuning across different platforms in the future, thereby truly simplifying the complexity of use for our users.

### Code and data availability

Our method's source code and documentation are available for download on GitHub (https://github.com/AI4EarthLab/CFIO2).

Additionally, all the codes and data related to the experiments in this paper can be found in the following sources:

1. All the codes including the data of GOMO case: https://doi.org/10.5281/zenodo.14581371 (Dong Wang, 2024).

2. "JAR55", the inputdata required for Licom3 case: https://doi.org/10.5281/zenodo.14580688 (Dong Wang, 2024).

3. "inputdata_licom", the inputdata required for Licom3 case: https://doi.org/10.5281/zenodo.14581279 (Dong Wang, 2024).

4. "input_data", the inputdata required for Licom3 case: https://doi.org/10.5281/zenodo.14581337 (Dong Wang, 2024).

The "Readme.docx" in Database 1 describes how to use this code and the associated data to reproduce the results presented in this article.

### Author contributions

Xiaomeng Huang and Dong Wang jointly conceived the idea for this paper. Dong Wang was responsible for the detailed planning of the algorithms for the methods presented, as well as for the development and testing of the code. Xiaomeng Huang

conducted a review and analysis of the code and data. Dong Wang completed the initial draft of the manuscript, while Xiaomeng Huang took on the role of reviewing and editing.



**Competing interests**

The author, Xiaomeng Huang, is a member of the editorial board of GMD.

**Acknowledgements.**

This work is supported by the National Key Research and Development Program of China (2022YFE0195900) the National Natural Science Foundation of China (42125503, 42430602)

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
