# Peer review of "Optimizing output operations in high-resolution climate models through dynamic scheduling"

_EGUsphere, 2024_

## Author Comment (AC1)

Dear editor and reviewer,

First of all, we would like to express our sincere appreciation to your valuable feed-backs. Your comments are highly insightful and enable us to substantially improve the quality of our manuscript. Below are our point-by-point responses to all the comments and our plans to revise the manuscript.

1. "This paper introduces an updated version of parallel I/O library for climate models. The authors have implemented new methods in data forwarding, data arrangement and data writing operations for improving the efficiency of parallel I/O. Moreover, a reinforcement learning scheme is provided to optimize timings for data arrangement processes. This is a nice piece of work that help optimizing I/O output operations in climate models. Although not much science on model results is provided, the architecture of parallel I/O operations is interesting. "

[Response]:

We sincerely thank the reviewer for their positive feedback on our work and their appreciation of the architecture of the parallel I/O operations. We agree with the reviewer that the current paper focuses primarily on the technical improvements and efficiency optimization of the parallel I/O library, rather than the scientific results of climate models. This is because the main goal of this work is to address the performance bottlenecks in I/O operations, which are critical for enabling large-scale climate simulations. Meanwhile, we believe that the improvement in I/O efficiency, which reduces model runtime, will enable climate model researchers to conduct more simulations or experiments within a limited time frame. This, in turn, will indirectly assist model researchers in discovering and addressing more scientific issues.

2. " Lines 205-210 and Figure 7: it is not explained in the manuscript that how could we make sure same data blocks are sent to a processor in data forwarding. Also, is the data rearrangement done within a processor or through a multi-processor task? Could you discuss how efficient is your rearrangement task for big datasets. Algorithm 1 in page 10: I suggest change this into a pseudo-code. Also, I suggest authors to discuss how efficient is this algorithm for metadata, particularly the send-receive communications."

[Response]:

We appreciate the reviewer's insightful comments and would like to clarify the implementation details regarding data forwarding and data rearrangement.

In the data forwarding phase, as illustrated in Figure 7, data is forwarded to Process 1 and Process 6. It is important to note that during this phase, there is no requirement for the same data blocks sent to a processor in data forwarding.

The data rearrangement process consists of two main steps:

● Data Exchange Between Processes: In this step, Process 1 and Process 6 exchange data blocks based on their respective `starts` and `counts`. After this exchange, both Process 1 and Process 6 will possess four data segments, which are geographically contiguous. This exchange is facilitated through MPI communication.

● Local Data Assembly: In this step, Process 1 and Process 6 independently assemble their respective four data segments into a larger, memory-contiguous block. This step is performed locally by each process without requiring further inter-process communication.

The efficiency of the data rearrangement process is notably high. As evidenced by the comparison between execution times for No. 5 and No. 6 in Table 4, the time spent on data rearrangement constitutes a minimal portion of the overall execution time. Furthermore, while the absolute time for data rearrangement may increase with larger datasets, its relative proportion to the total execution time remains largely unchanged. The same principle applies to the time spent on send-receive communications. To provide a more quantitative analysis, we will include additional experimental data in the next version of the manuscript.

We will explain the above more clearly in the manuscript. Regarding Algorithm 1 on page 10, we thank the reviewer for the suggestion to convert it into pseudo-code.

3.  Pages 11 and 12: How are trade-off 1 and 2 optimized? If they are not efficient, the I/O operation will not be improved.
[Response]:
Regarding the optimization of Trade-off 1 (the timing of file output operations), we have implemented two approaches:

- First, we provide an explicit output statement, similar to `MPI_Wait` in MPI, allowing users to explicitly specify when to perform output operations.
- Second, if users do not explicitly trigger the output, the system will automatically perform the output operation when the I/O processes are all occupied and can no longer buffer additional output variables. At this point, the buffer is cleared.

For Trade-off 2 (the number of I/O processes), based on our experimental experience, selecting between 4 to 16 processes as I/O processes has proven to be effective.

Both trade-offs fundamentally depend on the performance of the underlying hardware. This is why we introduced a reinforcement learning-based approach in this work. The method first collects performance data from the machine, and then an agent in a virtual environment automatically explores and identifies the optimal strategy.

We will explain the above more clearly in the manuscript.

4.  Also, the training process of reinforcement learning is not discussed. A fair comment here is that if you spend too much time on training your ML algorithm, that should be considered in evaluation of your updated I/O operation versus the previous I/O approaches.
[Response]:
We appreciate the reviewer's valuable feedback regarding the training process of the reinforcement learning (RL) algorithm. We will include a detailed discussion of the RL training process in the next version of the manuscript.

The reviewer raises a fair point: if the training process of the RL algorithm is too time-consuming, it should be factored into the evaluation of the updated I/O operations compared to previous approaches. In our experiments, using an Nvidia GeForce RTX 3090 GPU, the RL training process takes approximately 30 minutes.

Based on this, we recommend the following:

- If the simulation is not intended for long-term runs, the default parameter configuration in

our library already provides significant acceleration, and the RL-based approach may not be necessary.
- However, for simulations that require long-term execution, the benefits of using RL to explore and identify optimal strategies are substantial, justifying the additional training time.

We will ensure that this rationale is clearly articulated in the revised manuscript, along with a more detailed discussion of the RL training process and its implications.

5. Figure 1: there is empty space on the right side. I suggest moving panel (b) to top left side. This will save some space. Same comment is for Figure 2.
[Response]:
Thanks for the suggestion. We will reformat both figures in the revised manuscript.

We really appreciate your highly constructive comments.

Best wishes,
Dong Wang, Xiaomeng Huang

---

## Author Comment (AC2)

Dear editor and reviewer,

First of all, we would like to express our sincere appreciation for your valuable feedback. Your review is not only highly insightful but also extremely meticulous. You have provided us with many important suggestions, and you have also pointed out numerous formatting errors in detail. Your comments will help us to substantially improve the quality of our manuscript. Below are our point-by-point responses to all the comments and our plans to revise the manuscript.

**Major comments:**

1. Please clarify the reinforcement learning methodology. Provide detailed information on the RL agent's architecture (e.g., algorithm type, reward function specifics, training duration, and hyperparameters). Specify how performance data (network speed, PnetCDF rates) is collected and preprocessed for the virtual environment. It would be helpful to add pseudocode or a flowchart for the RL training process.

[Response]:

In this work, we employ the Proximal Policy Optimization (PPO) model for reinforcement learning. The reward function is defined as the total time taken to complete all I/O operations. The training was conducted on an NVIDIA 3090 GPU and lasted approximately 30 minutes. Specific hyper-parameters and technical details will be explicitly outlined in the revised version of the paper. Additionally, we will include pseudo-code and a flowchart to illustrate the RL training process for better clarity.

2. The evaluation system can be expanded by including metrics beyond speedup, such as CPU/memory utilization during I/O phases, network overhead, or buffer management efficiency.

[Response]:

Thank you for the suggestion. We will conduct additional experiments and include these metrics (e.g., CPU/memory utilization, network overhead, and buffer management efficiency) in the revised version to provide a more comprehensive evaluation.

3. Some discussions about trade-offs between resource consumption and performance gains can be added in the revised version.

[Response]:

We appreciate your suggestion. In the revised version, we will add a detailed discussion on the trade-offs between resource consumption and performance gains to provide deeper insights into the efficiency of our approach.

4. Please explicitly address limitations of CFIO2.0, such as dependency on pre-collected data for RL, scalability across heterogeneous file systems (e.g., Lustre vs. ParaStor), or adaptability to non-climate modeling workloads.

[Response]:

Thank you for pointing this out. We will include a discussion on the limitations of CFIO2.0 in the revised version. However, we would like to clarify that the dependency on pre-collected data for RL training is optional. Even without RL pre-training, the default parameters can still achieve satisfactory performance. Regarding file system support, since our work is built on PnetCDF for

file I/O, the underlying logic and strategies are algorithm-based and not limited by the file system. Therefore, our method theoretically supports any file system that PnetCDF supports, ensuring compatibility with most file systems. Additionally, our work is specifically designed for models that using NetCDF format file as output. Any MPI parallel program using this file format can be supported, while other formats or non-MPI programs are not within the scope of this study. We will explicitly state these points in the revised manuscript.

5. Ensure that all parameters (e.g., stride values in PIO experiments) are explicitly defined in tables or appendices.
[Response]:
Thank you for the suggestion. We will carefully review the manuscript and ensure that all parameters, including stride values in PIO experiments, are explicitly defined in tables or appendices.

**Minor suggestions:**
1. P1. Line 14-15: it is unclear whether the increases in ouput efficiency are across the two models or two I/O strategies.
[Response]:
The 1.54x speedup refers to the overall runtime improvement of the GOMO model when using our method compared to not using it. The 13.1x speedup refers to the overall runtime improvement of the LICOM model when using our method compared to not using it. We will clarify this in the revised version.

2. Page 3, Line 70: "Figure 1: Concurrent I/O and Computation." → Align with figure numbering (e.g., "Figure 1: (a) Alternating... (b) Parallel...").
[Response]:
We will correct this to align with the figure numbering.

3. Page 18, Line 385: "LICOM case study" → Add a colon for consistency ("LICOM Case Study:").
[Response]:
We will add a colon for consistency.

4. Figure 3 Caption (Page 4, Lines 100–110): Move the lengthy step-by-step description from the caption to the main text or a supplementary note to improve readability.
[Response]:
Thank you for the suggestion. We will move the detailed step-by-step description from the caption to the main text or a supplementary note to enhance readability.

5. Section 5.3 (Page 19, Lines 410–415): Label subfigures in Figure 13 as "Fig. 13a" and "Fig. 13b" instead of "(a)" and "(b)" to avoid confusion.
[Response]:
We will update the labels to "Fig. 13a" and "Fig. 13b" for clarity.

6. References (Page 23–24): Correct "Corbetty et al. (1996)" to "Corbett et al. (1996)" in the text (Page 2, Line 45).
[Response]:
We will correct this typo in the revised version.

7. Ensure all citations (e.g., "Kang et al. (2019)") have corresponding entries in the References section.
[Response]:
We will verify that all citations have corresponding entries in the References section.

8. Code Availability (Page 22, Lines 525–530): Verify that all Zenodo links are functional and datasets are publicly accessible.
[Response]:
We have verified that all Zenodo links are functional, and the datasets are publicly accessible.

We really appreciate your highly constructive comments.

Best wishes,
Dong Wang, Xiaomeng Huang